# Noncommutative $C^*$-algebra Net: Learning Neural Networks with Powerful Product Structure in $C^*$-algebra

## Abstract

We propose a new generalization of neural networks with noncommutative $C^*$-algebra. An important feature of $C^*$-algebras is their noncommutative structure of products, but the existing $C^*$-algebra net frameworks have only considered commutative $C^*$-algebras. We show that this noncommutative structure of $C^*$-algebras induces powerful effects in learning neural networks. Our framework has a wide range of applications, such as learning multiple related neural networks simultaneously with interactions and learning invariant features with respect to group actions. The validity of our framework numerically illustrates its potential power.

## 1 Introduction

Generalization of the parameter space of neural networks beyond real numbers brings intriguing possibilities. For instance, using complex numbers (Hirose, 1992; Nishikawa et al., 2005; Amin et al., 2008; Yadav et al., 2005; Trabelsi et al., 2018; Lee et al., 2022) or quaternion numbers (Nitta, 1995; Arena et al., 1997; Zhu et al., 2018; Gaudet & Maida, 2018) as neural network parameters is more intuitive and effective, particularly in signal processing, computer vision, and robotics domains. Clifford-algebra, the generalization of these numbers, allows more flexible geometrical data processing and is applied to neural networks to handle rich geometric relationships in data (Pearson & Bisset, 1994; Buchholz, 2005; Buchholz & Sommer, 2008; Rivera-Rovelo et al., 2010; Zang et al., 2022; Brandstetter et al., 2022; Ruhe et al., 2023b;a). Different from these approaches focusing on the geometric perspective of parameter values, an alternative direction of generalization is to use function-valued parameters (Rossi & Conan-Guez, 2005; Thind et al., 2023), broadening the applications of neural networks to functional data.

Hashimoto et al. (2022) proposed $C^*$-algebra net, which generalizes neural network parameters to (commutative) $C^*$-algebra — a generalization of complex numbers (Murphy, 1990; Hashimoto et al., 2021). They adopted continuous functions on a compact space as a commutative $C^*$-algebra and present a new interpretation of function-valued neural networks: infinitely many real-valued or complex-valued neural networks are continuously combined into a single $C^*$-algebra net. For example, networks for the same task with different training datasets or different initial parameters can be combined continuously, which enables efficient learning using shared information among infinite combined networks. Such interaction among networks is similar to learning from related tasks, such as ensemble learning (Dong et al., 2020; Ganaie et al., 2022) and multitask learning (Zhang & Yang, 2022). However, because the product structure in the $C^*$-algebra that Hashimoto et al. (2022) focuses on is commutative, such networks cannot take advantage of rich product structures of $C^*$-algebras and, instead, require specially designed loss functions to induce the necessary interaction.

To fully exploit rich product structures, we propose a new generalization of neural networks with noncommutative $C^*$-algebra. Typical examples of $C^*$-algebras include the space of diagonal matrices, which is commutative in terms of matrix product, and the space of squared matrices, which is noncommutative. Specifically, in the case of diagonal matrices, their product is simply computed by the multiplication of each diagonal element independently, and thus, they are commutative. On the other hand, the product of two squared nondiagonal matrices is the sum of the products between different elements, and each resultant di-

agonal element depends on other diagonal elements through nondiagonal elements, which can be interpreted as interaction among diagonal elements.

Such product structures derived from noncommutativity are powerful when used as neural network parameters. Keeping $C^*$-algebra over matrices as an example, a neural network with parameters of nondiagonal matrices can naturally induce interactions among multiple neural networks with real or complex-valued parameters by regarding each diagonal element as a parameter of such networks. Because interactions are encoded in the parameters, specially designed loss functions to induce interactions are unnecessary for such noncommutative $C^*$-algebra nets. This property clearly contrasts the proposed framework with the existing commutative $C^*$-algebra nets. Another example is a neural network with group $C^*$-algebra parameters, which is naturally group-equivariant without designing special network architectures.

Our main contributions in this paper are summarized as follows:

- We generalize the commutative $C^*$-algebra net proposed by Hashimoto et al. (2022) to noncommutative $C^*$-algebra, which can take advantage of the noncommutative product structure in the $C^*$-algebra when learning neural networks.

- We present two examples of the general noncommutative $C^*$-algebra net: $C^*$-algebra net over matrices and group $C^*$-algebra net. A $C^*$-algebra net over matrices can naturally combine multiple standard neural networks with interactions. Neural networks with group $C^*$-algebra parameters are naturally group-equivariant without modifying neural structures.

- Numerical experiments illustrate the validity of these noncommutative $C^*$-algebra nets, including interactions among neural networks.

We emphasize that $C^*$-algebra is a powerful tool for neural networks, and our work provides a lot of important perspectives about its application.

## 2 Background

In this section, we review the mathematical background of $C^*$-algebra required for this paper and the existing $C^*$-algebra net. For more theoretical details of the $C^*$-algebra, see, for example, Murphy (1990).

### 2.1 $C^*$-algebra

$C^*$-algebra is a generalization of the space of complex values. It has structures of the product, involution $^*$, and norm.

**Definition 1 ($C^*$-algebra)**   *A set $\mathcal{A}$ is called a $C^*$-algebra if it satisfies the following conditions:*

1. *$\mathcal{A}$ is an algebra over $\mathbb{C}$ and equipped with a bijection $(\cdot)^* : \mathcal{A} \to \mathcal{A}$ that satisfies the following conditions for $\alpha, \beta \in \mathbb{C}$ and $c, d \in \mathcal{A}$:*

   - *$(\alpha c + \beta d)^* = \overline{\alpha} c^* + \overline{\beta} d^*$,*
   - *$(cd)^* = d^* c^*$,*
   - *$(c^*)^* = c$.*

2. *$\mathcal{A}$ is a normed space with $\| \cdot \|$, and for $c, d \in \mathcal{A}$, $\|cd\| \leq \|c\| \, \|d\|$ holds. In addition, $\mathcal{A}$ is complete with respect to $\| \cdot \|$.*

3. *For $c \in \mathcal{A}$, $\|c^* c\| = \|c\|^2$ ($C^*$-property) holds.*

The product structure in $C^*$-algebras can be both commutative and noncommutative.

**Example 1 (Commutative $C^*$-algebra)** *Let $\mathcal{A}$ be the space of continuous functions on a compact Hausdorff space $\mathcal{Z}$. We can regard $\mathcal{A}$ as a $C^*$-algebra by setting*

- *Product: Pointwise product of two functions, i.e., for $a_1, a_2 \in \mathcal{A}$, $a_1 a_2(z) = a_1(z) a_2(z)$.*
- *Involution: Pointwise complex conjugate, i.e., for $a \in \mathcal{A}$, $a^*(z) = \overline{a(z)}$.*

- *Norm: Sup norm, i.e., for $a \in \mathcal{A}$, $\|a\| = \sup_{z \in \mathcal{Z}} |a(z)|$.*

*In this case, the product in $\mathcal{A}$ is commutative.*

**Example 2 (Noncommutative $C^*$-algebra)** *Let $\mathcal{A}$ be the space of bounded linear operators on a Hilbert space $\mathcal{H}$, which is denoted by $\mathcal{B}(\mathcal{H})$. We can regard $\mathcal{A}$ as a $C^*$-algebra by setting*

- *Product: Composition of two operators,*

- *Involution: Adjoint of an operator,*

- *Norm: Operator norm of an operator, i.e., for $a \in \mathcal{A}$, $\|a\| = \sup_{v \in \mathcal{H}, \|v\|_{\mathcal{H}} = 1} \|av\|_{\mathcal{H}}$.*

*Here, $\| \cdot \|_{\mathcal{H}}$ is the norm in $\mathcal{H}$. In this case, the product in $\mathcal{A}$ is noncommutative. Note that if $\mathcal{H}$ is a $d$-dimensional space for a finite natural number $d$, then elements in $\mathcal{A}$ are $d$ by $d$ matrices.*

**Example 3 (Group $C^*$-algebra)** *The group $C^*$-algebra on a group $G$, which is denoted as $C^*(G)$, is the set of maps from $G$ to $\mathbb{C}$ equipped with the following product, involution, and norm:*

- *Product: $(a \cdot b)(g) = \int_G a(h)b(h^{-1}g)\mathrm{d}\lambda(h)$ for $g \in G$,*

- *Involution: $a^*(g) = \Delta(g^{-1})\overline{a(g^{-1})}$ for $g \in G$,*

- *Norm: $\|a\| = \sup_{[\pi] \in \hat{G}} \|\pi(a)\|$,*

*where $\Delta(g)$ is a positive number satisfying $\lambda(Eg) = \Delta(g)\lambda(E)$ for the Haar measure $\lambda$ on $G$. In addition, $\hat{G}$ is the set of equivalence classes of irreducible unitary representations of $G$. Note that if $G$ is discrete, then $\lambda$ is the counting measure on $G$. In this paper, we focus mainly on the product structure of $C^*(G)$. For details of the Haar measure and representations of groups, see Kirillov (1976). If $G = \mathbb{Z}/p\mathbb{Z}$, then $C^*(G)$ is $C^*$-isomorphic to the $C^*$-algebra of circulant matrices (Hashimoto et al., 2023). Note also that if $G$ is noncommutative, then $C^*(G)$ can also be noncommutative.*

### 2.2 $C^*$-algebra net

Hashimoto et al. (2022) proposed generalizing real-valued neural network parameters to commutative $C^*$-algebra-valued ones, which enables us to represent multiple real-valued models as a single $C^*$-algebra net. Here, we briefly review the existing (commutative) $C^*$-algebra net. Let $\mathcal{A} = C(\mathcal{Z})$, the commutative $C^*$-algebra of continuous functions on a compact Hausdorff space $\mathcal{Z}$. Let $H$ be the depth of the network and $N_0, \ldots, N_H$ be the width of each layer. For $i = 1, \ldots, H$, set $W_i : \mathcal{A}^{N_{i-1}} \to \mathcal{A}^{N_i}$ as an Affine transformation defined with an $N_i \times N_{i-1}$ $\mathcal{A}$-valued matrix and an $\mathcal{A}$-valued bias vector in $\mathcal{A}^{N_i}$. In addition, set a nonlinear activation function $\sigma_i : \mathcal{A}^{N_i} \to \mathcal{A}^{N_i}$. The commutative $C^*$-algebra net $f : \mathcal{A}^{N_0} \to \mathcal{A}^{N_H}$ is defined as

$$f = \sigma_H \circ W_H \circ \cdots \circ \sigma_1 \circ W_1. \tag{1}$$

By generalizing neural network parameters to functions, we can combine multiple standard (real-valued) neural networks continuously, which enables them to learn efficiently. In this paper, each real-valued network in a $C^*$-algebra net is called sub-model. We show an example of commutative $C^*$-algebra nets below. To simplify the notation, we focus on the case where the network does not have biases. However, the same arguments are valid for the case where the network has biases.

#### 2.2.1 The case of diagonal matrices

If $\mathcal{Z}$ is a finite set, then $\mathcal{A} = \{a \in \mathbb{C}^{d \times d} \mid a \text{ is a diagonal matrix}\}$. The $C^*$-algebra net $f$ on $\mathcal{A}$ corresponds to $d$ separate real or complex-valued sub-models. In the case of $\mathcal{A} = C(\mathcal{Z})$, we can consider that infinitely many networks are continuously combined, and the $C^*$-algebra net $f$ with diagonal matrices is a discretization of the $C^*$-algebra net over $C(\mathcal{Z})$. Indeed, denote by $x^j$ the vector composed of the $j$th diagonal elements of $x \in \mathcal{A}^N$, which is defined as the vector in $\mathbb{C}^N$ whose $k$th element is the $j$th diagonal element of the $\mathcal{A}$-valued $k$th element of $x$. Assume the activation function $\sigma_i : \mathcal{A}^N \to \mathcal{A}^N$ is defined as $\sigma_i(x)^j = \tilde{\sigma}_i(x^j)$ for some $\tilde{\sigma}_i : \mathbb{C}^N \to \mathbb{C}^N$. Since the $j$th diagonal element of $a_1 a_2$ for $a_1, a_2 \in \mathcal{A}$ is the product of the $j$th element of $a_1$

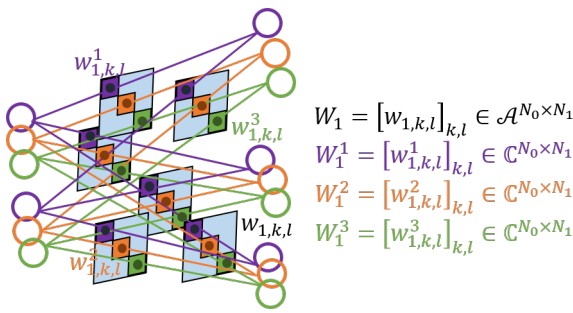

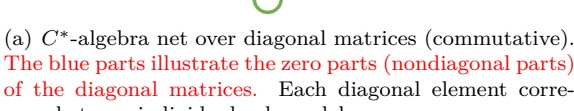

$$W_1 = \left[w_{1,k,l}\right]_{k,l} \in \mathcal{A}^{N_0 \times N_1}$$
$$W_1^1 = \left[w_{1,k,l}^1\right]_{k,l} \in \mathbb{C}^{N_0 \times N_1}$$
$$W_1^2 = \left[w_{1,k,l}^2\right]_{k,l} \in \mathbb{C}^{N_0 \times N_1}$$
$$W_1^3 = \left[w_{1,k,l}^3\right]_{k,l} \in \mathbb{C}^{N_0 \times N_1}$$

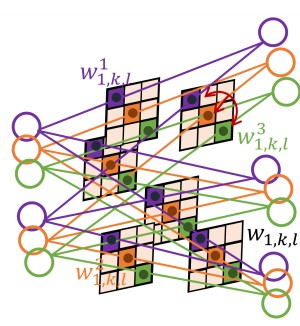

(a) $C^*$-algebra net over diagonal matrices (commutative). The blue parts illustrate the zero parts (nondiagonal parts) of the diagonal matrices. Each diagonal element corresponds to an individual sub-model.

(b) $C^*$-algebra net over nondiagonal matrices (noncommutative). Unlike the case of diagonal matrices, nondiagonal parts (colored in orange) are not zero. The nondiagonal elements induce the interactions among multiple sub-models.

Figure 1: Difference between commutative and noncommutative $C^*$-algebra nets from the perspective of interactions among sub-models.

and $a_2$, we have

$$f(x)^j = \tilde{\sigma}_H \circ W_H^j \circ \cdots \circ \tilde{\sigma}_1 \circ W_1^j, \tag{2}$$

where $W_i^j \in \mathbb{C}^{N_i \times N_{i-1}}$ is the matrix whose $(k, l)$-entry is the $j$th diagonal of the $(k, l)$-entry of $W_i \in \mathcal{A}^{N_i \times N_{i-1}}$. Figure 1 (a) schematically shows the $C^*$-algebra net over diagonal matrices.

## 3 Noncommutative $C^*$-algebra Net

Although the existing $C^*$-algebra net provides a framework for applying $C^*$-algebra to neural networks, it focuses on commutative $C^*$-algebras, whose product structure is simple. Therefore, we generalize the existing commutative $C^*$-algebra net to noncommutative $C^*$-algebra. Since the product structures in noncommutative $C^*$-algebras are more complicated than those in commutative $C^*$-algebras, they enable neural networks to learn features of data more efficiently. For example, if we focus on the $C^*$-algebra of matrices, then the neural network parameters describe interactions between multiple real-valued sub-models (see Section 3.1.1).

Let $\mathcal{A}$ be a general $C^*$-algebra and consider the network $f$ in the same form as Equation (1). We emphasize that in our framework, the choice of $\mathcal{A}$ is not restricted to a commutative $C^*$-algebra. We list examples of $\mathcal{A}$ and their validity for learning neural networks below.

### 3.1 Examples of $C^*$-algebras for neural networks

Through showing several examples of $C^*$-algebras, we show that $C^*$-algebra net is a flexible and unified framework that incorporates $C^*$-algebra into neural networks. As mentioned in the previous section, we focus on the case where the network does not have biases for simplification in this subsection.

#### 3.1.1 Nondiagonal matrices

Let $\mathcal{A} = \mathbb{C}^{d \times d}$. Note that $\mathcal{A}$ is a noncommutative $C^*$-algebra. Of course, it is possible to consider matrix data, but we focus on the perspective of interaction among sub-models following Section 2.2. In this case, unlike the network (2), the $j$th diagonal element of $a_1 a_2 a_3$ for $a_1, a_2, a_3 \in \mathcal{A}$ depends not only on the $j$th diagonal element of $a_2$, but also the other diagonal elements of $a_2$. Thus, $f(x)^j$ depends not only on the sub-model corresponding to $j$th diagonal element discussed in Section 2.2.1, but also on other sub-models. The nondiagonal elements in $\mathcal{A}$ induce interactions between $d$ real or complex-valued sub-models.

Interaction among sub-models could be related to decentralized peer-to-peer machine learning Vanhaesebrouck et al. (2017); Bellet et al. (2018), where each agent learns without sharing data with others, while

improving its ability by leveraging other agents' information through communication. In our case, an agent corresponds to a sub-model, and communication is achieved by interaction. We will see the effect of interaction by the nondiagonal elements in $\mathcal{A}$ numerically in Section 4.1. Figure 1 (b) schematically shows the $C^*$-algebra net over nondiagonal matrices.

### 3.1.2 Block diagonal matrices

Let $\mathcal{A} = \{a \in \mathbb{C}^{d \times d} \mid a = \mathrm{diag}(\mathbf{a}_1, \ldots, \mathbf{a}_m),\ \mathbf{a}_i \in \mathbb{C}^{d_i \times d_i}\}$. The product of two block diagonal matrices $a = \mathrm{diag}(\mathbf{a}_1, \ldots, \mathbf{a}_m)$ and $b = \mathrm{diag}(\mathbf{b}_1, \ldots, \mathbf{b}_m)$ can be written as

$$ab = \mathrm{diag}(\mathbf{a}_1\mathbf{b}_1, \ldots, \mathbf{a}_m\mathbf{b}_m).$$

In a similar manner to Section 2.2.1, we denote by $\mathbf{x}^j$ the $N$ by $d_j$ matrix composed of the $j$th diagonal blocks of $x \in \mathcal{A}^N$. Assume the activation function $\sigma_i : \mathcal{A}^N \to \mathcal{A}^N$ is defined as $\sigma_i(x) = \mathrm{diag}(\tilde{\boldsymbol{\sigma}}_i^1(\mathbf{x}^1), \ldots, \tilde{\boldsymbol{\sigma}}_i^m(\mathbf{x}^m))$ for some $\tilde{\boldsymbol{\sigma}}_{i,j} : \mathbb{C}^{N \times d_j} \to \mathbb{C}^{N \times d_j}$. Then, we have

$$\mathbf{f}(\mathbf{x})^j = \tilde{\boldsymbol{\sigma}}_H^j \circ \mathbf{W}_H^j \circ \cdots \circ \tilde{\boldsymbol{\sigma}}_1^j \circ \mathbf{W}_1^j, \tag{3}$$

where $\mathbf{W}_i^j \in (\mathbb{C}^{d_j \times d_j})^{N_i \times N_{i-1}}$ is the block matrix whose $(k,l)$-entry is the $j$th block diagonal of the $(k,l)$-entry of $W_i \in \mathcal{A}^{N_i \times N_{i-1}}$. In this case, we have $m$ groups of sub-models, each of which is composed of interacting $d_j$ sub-models mentioned in Section 3.1.1. Indeed, the block diagonal case generalizes the diagonal and nondiagonal cases stated in Sections 2.2.1 and 3.1.1. If $d_j = 1$ for all $j = 1, \ldots, m$, then the network (3) is reduced to the network (2) with diagonal matrices. If $m = 1$ and $d_1 = d$, then the network (3) is reduced to the network with $d$ by $d$ nondiagonal matrices.

### 3.1.3 Circulant matrices

Let $\mathcal{A} = \{a \in \mathbb{C}^{d \times d} \mid a \text{ is a circulant matrix}\}$. Here, a circulant matrix $a$ is the matrix represented as

$$a = \begin{bmatrix} a_1 & a_d & \cdots & a_2 \\ a_2 & a_1 & \cdots & a_3 \\ & \ddots & \ddots & \\ a_d & a_{d-1} & \cdots & a_1 \end{bmatrix}$$

for $a_1, \ldots, a_d \in \mathbb{C}$. Note that in this case, $\mathcal{A}$ is commutative. Circulant matrices are diagonalized by the discrete Fourier matrix as follows (Davis, 1979). We denote by $F$ the discrete Fourier transform matrix, whose $(i,j)$-entry is $\omega^{(i-1)(j-1)}/\sqrt{p}$, where $\omega = \mathrm{e}^{2\pi\sqrt{-1}/d}$.

**Lemma 1** *Any circulant matrix $a$ is decomposed as $a = F\Lambda_a F^*$, where*

$$\Lambda_a = \mathrm{diag}\left( \sum_{i=1}^d a_i \omega^{(i-1)\cdot 0}, \ldots, \sum_{i=1}^d a_i \omega^{(i-1)(d-1)} \right).$$

Since $ab = F\Lambda_a\Lambda_b F^*$ for $a, b \in \mathcal{A}$, the product of $a$ and $b$ corresponds to the multiplication of each Fourier component of $a$ and $b$. Assume the activation function $\sigma_i : \mathcal{A}^N \to \mathcal{A}^N$ is defined such that $(F^*\sigma_i(x)F)^j$ equals to $\hat{\tilde{\sigma}}_i((FxF^*)^j)$ for some $\hat{\tilde{\sigma}}_i : \mathbb{C}^N \to \mathbb{C}^N$. Then, we obtain the network

$$(F^*f(x)F)^j = \hat{\tilde{\sigma}}_H \circ \hat{W}_H^j \circ \cdots \circ \hat{\tilde{\sigma}}_1 \circ \hat{W}_1^j, \tag{4}$$

where $\hat{W}_i^j \in \mathbb{C}^{N_i \times N_{i-1}}$ is the matrix whose $(k,l)$-entry is $(Fw_{i,k,l}F^*)^j$, where $w_{i,k,l}$ is the the $(k,l)$-entry of $W_i \in \mathcal{A}^{N_i \times N_{i-1}}$. The $j$th sub-model of the network (4) corresponds to the network of the $j$th Fourier component.

**Remark 1** *The $j$th sub-model of the network (4) does not interact with those of other Fourier components than the $j$th component. This fact corresponds to the fact that $\mathcal{A}$ is commutative in this case. Analogous to the case in Section 3.1.1, if we set $\mathcal{A}$ as noncircular matrices, then we obtain interactions between sub-models corresponding to different Fourier components.*

### 3.1.4 Group $C^*$-algebra on a symmetric group

Let $G$ be the symmetric group on the set $\{1, \ldots, d\}$ and let $\mathcal{A} = C^*(G)$. Note that since $G$ is noncommutative, $C^*(G)$ is also noncommutative. Then, the output $f(x) \in \mathcal{A}^{N_H}$ is the $\mathbb{C}^{N_H}$-valued map on $G$. Using the product structure introduced in Example 3, we can construct a network that takes the permutation of data into account. Indeed, an element $w \in \mathcal{A}$ of a weight matrix $W \in \mathcal{A}^{N_{i-1} \times N_i}$ is a function on $G$. Thus, $w(g)$ describes the weight corresponding to the permutation $g \in G$. Since the product of $x \in C^*(G)$ and $w$ is defined as $wx(g) = \sum_{h \in G} w(h)x(h^{-1}g)$, by applying $W$, all the weights corresponding to the permutations affect the input. For example, let $z \in \mathbb{R}^d$ and set $x \in C^*(G)$ as $x(g) = g \cdot z$, where $g \cdot z$ is the action of $g$ on $z$, i.e., the permutation of $z$ with respect to $g$. Then, we can input all the patterns of permutations of $z$ simultaneously, and by virtue of the product structure in $C^*(G)$, the network is learned with the interaction among these permutations. Regarding the output, if the network is learned so that the outputs $y$ become constant functions on $G$, i.e., $y(g) = c$ for some constant $c$, then it means that $c$ is invariant with respect to $g$, i.e., invariant with respect to the permutation. We will numerically investigate the application of the group $C^*$-algebra net to permutation invariant problems in Section 4.2.

**Remark 2** *If the activation function $\sigma$ is defined as $\sigma(x)(g) = \sigma(x(g))$, i.e., applied elementwisely to $x$, then the network $f$ is permutation equivariant. That is, even if the input $x(g)$ is replaced by $x(gh)$ for some $h \in G$, the output $f(x)(g)$ is replaced by $f(x)(gh)$. This is because the product in $C^*(G)$ is defined as a convolution. This feature of the convolution has been studied for group equivariant neural networks (Lenssen et al., 2018; Cohen et al., 2019; Sannai et al., 2021; Sonoda et al., 2022). The above setting of the $C^*$-algebra net provides us with a design of group equivariant networks from the perspective of $C^*$-algebra.*

**Remark 3** *Since the number of elements of $G$ is $d!$, elements in $C^*(G)$, which are functions on $G$, are represented as $d!$-dimensional vectors. For the case where $d$ is large, we need a method for efficient computations, which is future work.*

### 3.1.5 Bounded linear operators on a Hilbert space

For functional data, we can also set $\mathcal{A}$ as an infinite-dimensional space. Using infinite-dimensional $C^*$-algebra for analyzing functional data has been proposed (Hashimoto et al., 2021). We can also adopt this idea for neural networks. Let $\mathcal{A} = \mathcal{B}(L^2(\Omega))$ for a measure space $\Omega$. Set $\mathcal{A}_0 = \{a \in \mathcal{A} \mid a \text{ is a multiplication operator}\}$. Here, a multiplication operator $a$ is a linear operator that is defined as $av = v \cdot u$ for some $u \in L^\infty(\Omega)$. The space $\mathcal{A}_0$ is a generalization of the space of diagonal matrices to the infinite-dimensional space. If we restrict elements of weight matrices to $\mathcal{A}_0$, then we obtain infinitely many sub-models without interactions. Since outputs are in $\mathcal{A}_0^{N_H}$, we can obtain functional data as outputs. Similar to the case of matrices (see Section 3.1.1), by setting elements of weight matrices as elements in $\mathcal{A}$, we can take advantage of interactions among infinitely many sub-models.

## 3.2 Approximation of functions with interactions by $C^*$-algebra net

We observe what kind of functions the $C^*$-algebra net can approximate. In this subsection, we show the universality of $C^*$-algebra nets, which describes the representation power of models. We focus on the case of $\mathcal{A} = \mathbb{C}^{d \times d}$. Consider a shallow network $f : \mathcal{A}^{N_0} \to \mathcal{A}$ defined as $f(x) = W_2^* \sigma(W_1 x + b)$, where $W_1 \in \mathcal{A}^{N_1 \times N_0}$, $W_2 \in \mathcal{A}^{N_1}$, and $b \in \mathcal{A}^{N_1}$. Let $\tilde{f} : \mathcal{A}^{N_0} \to \mathcal{A}$ be the function in the form of $\tilde{f}(x) = [\sum_{j=1}^d f_{kj}(x^l)]_{kl}$, where $f_{kj} : \mathbb{C}^{N_0 d} \to \mathbb{R}$. Here, we abuse the notation and denote by $x^l \in \mathbb{C}^{N_0 d}$ the $l$th column of $x$ regarded as an $N_0 d$ by $d$ matrix. Assume $f_{kj}$ is represented as

$$f_{kj}(x) = \int_{\mathbb{R}} \int_{\mathbb{R}^{N_0 d}} T_{kj}(w, b) \sigma(w^* x + b) \mathrm{d}w \, \mathrm{d}b \tag{5}$$

for some $T_{kj} : \mathbb{R}^{N_0 d} \times \mathbb{R} \to \mathbb{R}$. By the theory of the ridgelet transform, such $T_{kj}$ exists for most realistic settings (Candès, 1999; Sonoda & Murata, 2017). For example, Sonoda & Murata (2017) showed the following result.

**Proposition 1** *Let $\mathcal{S}$ be the space of rapidly decreasing functions on $\mathbb{R}$ and $\mathcal{S}_0'$ be the dual space of the Lizorkin distribution space on $\mathbb{R}$. Assume a function $\tilde{f}$ has the form of $\tilde{f}(x) = [\sum_{j=1}^d f_{kj}(x^l)]_{kl}$, and $f_{kj}$ and $\hat{f_{kj}}$ are in $L^1(\mathbb{R}^{N_0 d})$, where $\hat{f}$ represents the Fourier transform of a function $f$. Assume in addition, $\sigma$ is in $\mathcal{S}_0'$, and there exists $\psi \in \mathcal{S}$ such that $\int_{\mathbb{R}} \overline{\hat{\psi}(x)} \hat{\sigma}(x)/|x| \mathrm{d}x$ is nonzero and finite. Then, there exists $T_{kj} : \mathbb{R}^{N_0 d} \times \mathbb{R} \to \mathbb{R}$ such that $f_{kj}$ admits a representation of Equation (5).*

Here, the Lizorkin distribution space is defined as $\mathcal{S}_0 = \{\phi \in \mathcal{S} \mid \int_{\mathbb{R}} x^k \phi(x) \mathrm{d}x = 0 \; k \in \mathbb{N}\}$. Note that the ReLU is in $S_0'$. We discretize Equation (5) by replacing the Lebesgue measures with $\sum_{i=1}^{N_1} \delta_{w_{ij}}$ and $\sum_{i=1}^{N_1} \delta_{b_{ij}}$, where $\delta_w$ is the Dirac measure centered at $w$. Then, the $(k,l)$-entry of $\tilde{f}(x)$ is written as

$$\sum_{j=1}^d \sum_{i=1}^{N_1} T_{kj}(w_{ij}, b_{ij}) \sigma(w_{ij}^* x^l + b_{ij}).$$

Setting the $i$-th element of $W_2 \in \mathcal{A}^{N_1}$ as $[T_{kj}(w_{ij}, b_{ij})]_{kj}$, the $(i,m)$-entry of $W_1 \in \mathcal{A}^{N_1 \times N_0}$ as $[(w_{i,j})_{md+l}]_{jl}$, the $i$th element of $b \in \mathcal{A}^{N_1}$ as $[b_j]_{jl}$, we obtain

$$\sum_{j=1}^d \sum_{i=1}^{N_1} T_{kj}(w_{ij}, b_{ij}) \sigma(w_{ij}^* x^l + b_{ij}) = (W_2^k)^* \sigma(W_1 x^l + b^l),$$

which is the $(k,l)$-entry of $f(x)$. As a result, any function in the form of $\tilde{f}(x) = [\sum_{j=1}^d f_{kj}(x^l)]_{kl}$ with the assumption in Proposition 1 can be represented as a $C^*$-algebra net.

**Remark 4** *As we discussed in Sections 2.2.1 and 3.1.1, a $C^*$-algebra net over matrices can be regarded as $d$ interacting sub-models. The above argument insists the $l$th column of $f(x)$ and $\tilde{f}(x)$ depends only on $x^l$. Thus, in this case, if we input data $x^l$ corresponding to the $l$th sub-model, then the output is obtained as the $l$th column of the $\mathcal{A}$-valued output $f(x)$. On the other hand, the weight matrices $W_1$ and $W_2$ and the bias $b$ are used commonly in providing the outputs for any sub-model, i.e., $W_1$, $W_2$, and $b$ are learned using data corresponding to all the sub-models. Therefore, $W_1$, $W_2$, and $b$ induce interactions among the sub-models.*

## 4 Experiments

In this section, we numerically demonstrate the abilities of noncommutative $C^*$-algebra nets using nondiagonal $C^*$-algebra nets over matrices in light of interaction and group $C^*$-algebra nets of its equivariant property. We use $C^*$-algebra-valued multi-layered perceptrons (MLPs) to simplify the experiments. However, they can be naturally extended to other neural networks, such as convolutional neural networks.

The models were implemented with `JAX` (Bradbury et al., 2018). Experiments were conducted on an AMD EPYC 7543 CPU and an NVIDIA A-100 GPU. See Appendix A.1 for additional information on experiments.

### 4.1 $C^*$-algebra nets over matrices

In a noncommutative $C^*$-algebra net over matrices consisting of nondiagonal-matrix parameters, each sub-model is expected to interact with others and thus improve performance compared with its commutative counterpart consisting of diagonal matrices. We demonstrate the effectiveness of such interaction using image classification and neural implicit representation (NIR) tasks in a similar setting with peer-to-peer learning such that data are separated for each submodel.

See Section 3.1.1 for the notations. When training the $j$th sub-model ($j = 1, 2, \ldots, d$), an original $N_0$-dimensional input data point $\boldsymbol{x} = [\boldsymbol{x}_1, \ldots, \boldsymbol{x}_{N_0}] \in \mathbb{R}^{N_0}$ is converted to its corresponding representation $x \in \mathcal{A}^{N_0} = \mathbb{R}^{N_0 \times d \times d}$ such that $x_{i,j,j} = \boldsymbol{x}_i$ for $i = 1, 2, \ldots, N_0$ and 0 otherwise. The loss to its $N_H$-dimensional output of a $C^*$-algebra net $y \in \mathcal{A}^{N_H}$ and the target $t \in \mathcal{A}^{N_H}$ is computed as $\ell(y_{:,j,j}, t_{:,j,j}) + \frac{1}{2} \sum_{k,(l \neq j)} (y_{k,j,l}^2 + y_{k,l,j}^2)$, where $\ell$ is a certain loss function; we use the mean squared error (MSE) for image classification and the Huber loss for NIR. The second and third terms suppress the nondiagonal elements of the outputs to 0. In both examples, we use leaky-ReLU as an activation function and apply it only to the diagonal elements of pre-activations.

### 4.1.1 Image classification

We conduct experiments of image classification tasks using MNIST (Le Cun et al., 1998), Kuzushiji-MNIST (Clanuwat et al., 2018), and Fashion-MNIST (Xiao et al., 2017), which are composed of 10-class $28 \times 28$ gray-scale images. Each sub-model is trained on a mutually exclusive subset sampled from the original training data and then evaluated on the entire test data. Each subset is sampled to be balanced, i.e., each class has the same number of training samples. As a baseline, we use a commutative $C^*$-algebra net over diagonal matrices, which consists of the same sub-models but cannot interact with other sub-models. Both noncommutative and commutative models share hyperparameters: the number of layers was set to 4, the hidden size was set to 128, and the models were trained for 30 epochs.

Table 1 shows average test accuracy. Accuracy can be reported in two distinct manners: the first approach averages the accuracy across individual sub-models ("Average"), and the other is to ensemble the logits of sub-models and then compute the accuracy ("Ensemble"). As can be seen, the noncommutative $C^*$-algebra net consistently outperforms its commutative counterpart, which is significant, particularly when the number of sub-models is 40. Note that when the number of sub-models is 40, the size of the training dataset for each sub-model is 40 times smaller than the original one, and thus, the commutative $C^*$-algebra net fails to learn. Nevertheless, the noncommutative $C^*$-algebra net retains performance mostly. Commutative $C^*$-algebra net improves performance by ensembling, but it achieves inferior performance to both Average and Ensemble noncommutative $C^*$-algebra net when the number of sub-models is larger than five. Such a performance improvement would be attributed to the fact that noncommutative models have more trainable parameters than commutative ones. Thus, we additionally compare commutative $C^*$-algebra net with a width of 128 and noncommutative $C^*$-algebra net with a width of 8, which have the same number of learnable parameters, when the total number of training data is set to 5000, ten times smaller than the experiments of Table 1. As seen in Table 2, while the commutative $C^*$-algebra net mostly fails to learn, the noncommutative $C^*$-algebra net learns successfully. These results suggest that the performance of noncommutative $C^*$-algebra net cannot solely be explained by the number of learnable parameters: the interaction among sub-models provides essential capability.

Furthermore, Table 3 illustrates that related tasks help performance improvement through interaction. Specifically, we prepare five sub-models per dataset, one of MNIST, Kuzushiji-MNIST, and Fashion-MNIST, and train a single (non)commutative $C^*$-algebra net consisting of 15 sub-models simultaneously. In addition to the commutative $C^*$-algebra net, where sub-models have no interaction, and the noncommutative $C^*$-algebra net, where each sub-model can interact with any other sub-models, we use a block-diagonal noncommutative $C^*$-algebra net (see Section 3.1.2), where each sub-model can only interact with other sub-models trained on the same dataset. Table 3 shows that the fully noncommutative $C^*$-algebra net surpasses the block-diagonal one on Kuzushiji-MNIST and Fashion-MNIST, implying that not only intra-task interaction but also inter-task interaction helps performance gain. Note these results are not directly comparable with the values of Tables 1 and 3, due to dataset subsampling to balance class sizes (matching MNIST's smallest class).

### 4.1.2 Neural implicit representation

In the next experiment, we use a $C^*$-algebra net over matrices to learn implicit representations of 2D images that map each pixel coordinate to its RGB colors (Sitzmann et al., 2020; Xie et al., 2022). Specifically, an input coordinate in $[0,1]^2$ is transformed into a random Fourier feature in $[-1,1]^{320}$ and then converted into its $C^*$-algebraic representation over matrices as an input to a $C^*$-algebra net over matrices. Similar to the image classification task, we compare noncommutative NIRs with commutative NIRs, using the following hyperparameters: the number of layers to 6 and the hidden dimension to 256. These NIRs learn $128 \times 128$-pixel images of ukiyo-e pictures from The Metropolitan Museum of Art[1] and photographs of cats from the AFHQ dataset (Choi et al., 2020).

Figure 2 (top) shows the curves of the average PSNR (Peak Signal-to-Noise Ratio) of sub-models corresponding to the image below. Both commutative and noncommutative $C^*$-algebra nets consist of five sub-models trained on five ukiyo-e pictures (see also Figure 6). PSNR, the quality measure, of the noncommutative NIR

---

[1] https://www.metmuseum.org/art/the-collection

Table 1: Average test accuracy of commutative and noncommutative $C^*$-algebra nets over matrices on test datasets. "Average" reports the average accuracy of sub-models, and "Ensemble" ensembles the logits of sub-models to compute accuracy. Interactions between sub-models that the noncommutative $C^*$-algebra net improves performance significantly when the number of sub-models is 40.

| Dataset | # sub-models | Commutative $C^*$-algebra nets | | Noncommutative $C^*$-algebra nets | |
| --- | --- | --- | --- | --- | --- |
| | | Average | Ensemble | Average | Ensemble |
| MNIST | 5 | $0.963 \pm 0.003$ | $0.970 \pm 0.001$ | $0.970 \pm 0.002$ | $0.976 \pm 0.001$ |
| | 10 | $0.937 \pm 0.004$ | $0.950 \pm 0.000$ | $0.956 \pm 0.002$ | $0.969 \pm 0.000$ |
| | 20 | $0.898 \pm 0.007$ | $0.914 \pm 0.002$ | $0.937 \pm 0.002$ | $0.957 \pm 0.001$ |
| | 40 | $0.605 \pm 0.007$ | $0.795 \pm 0.010$ | $0.906 \pm 0.004$ | $0.938 \pm 0.001$ |
| Kuzushiji-MNIST | 5 | $0.837 \pm 0.003$ | $0.871 \pm 0.001$ | $0.859 \pm 0.003$ | $0.888 \pm 0.002$ |
| | 10 | $0.766 \pm 0.008$ | $0.793 \pm 0.004$ | $0.815 \pm 0.007$ | $0.859 \pm 0.002$ |
| | 20 | $0.674 \pm 0.011$ | $0.710 \pm 0.001$ | $0.758 \pm 0.007$ | $0.817 \pm 0.001$ |
| | 40 | $0.453 \pm 0.026$ | $0.532 \pm 0.004$ | $0.682 \pm 0.008$ | $0.767 \pm 0.001$ |
| Fashion-MNIST | 5 | $0.862 \pm 0.001$ | $0.873 \pm 0.001$ | $0.868 \pm 0.002$ | $0.882 \pm 0.001$ |
| | 10 | $0.839 \pm 0.003$ | $0.850 \pm 0.001$ | $0.852 \pm 0.004$ | $0.871 \pm 0.001$ |
| | 20 | $0.790 \pm 0.010$ | $0.796 \pm 0.002$ | $0.832 \pm 0.005$ | $0.858 \pm 0.000$ |
| | 40 | $0.650 \pm 0.018$ | $0.674 \pm 0.001$ | $0.810 \pm 0.005$ | $0.841 \pm 0.000$ |

Table 2: Average test accuracy commutative and noncommutative $C^*$-algebra nets over matrices trained on 5000 data points with 20 sub-models. The width of the noncommutative model is set to 8 so that the number of learnable parameters is matched with its commutative counterpart.

| Dataset | Commutative | Noncommutative |
| --- | --- | --- |
| MNIST | $0.155 \pm 0.04$ | $0.779 \pm 0.02$ |
| Kuzushiji-MNIST | $0.140 \pm 0.03$ | $0.486 \pm 0.02$ |
| Fashion-MNIST | $0.308 \pm 0.05$ | $0.673 \pm 0.02$ |

grows faster, and correspondingly, it learns the details of ground truth images faster than its commutative version (Figure 2 bottom). Noticeably, the noncommutative representations reproduce colors even at the early stage of learning, although the commutative ones remain monochrome after 500 iterations of training. Along with the similar trends observed in the pictures of cats (Figure 3), these results further emphasize the effectiveness of the interaction. Longer-term results are presented in Figure 7.

This INR for 2D images can be extended to represent 3D models. Figure 4 shows synthesized views of 3D implicit representations using the same $C^*$-algebra MLPs trained on three 3D chairs from the ShapeNet dataset (Chang et al., 2015). The presented poses are unseen during training. Again, the noncommutative INR reconstructs the chair models with less noisy artifacts, indicating that interaction helps efficient learning. See Appendices A.1 and A.2 for details and results.

## 4.2 Group $C^*$-algebra nets

As another experimental example of $C^*$-algebra nets, we showcase group $C^*$-algebra nets, which we introduced in Section 3.1.4. The group $C^*$-algebra nets take functions on a symmetric group as input and return functions on the group as output.

Refer to Section 3.1.4 for notations. A group $C^*$-algebra net is trained on data $\{(x, y) \in \mathcal{A}^{N_0} \times \mathcal{A}^{N_H}\}$, where $x$ and $y$ are $N_0$- and $N_H$-dimensional vector-valued functions. Practically, such functions may be represented as real tensors, e.g., $x \in \mathbb{C}^{N_0 \times \#G}$, where $\#G$ is the size of $G$. Using product between functions explained in Section 3.1.4 and element-wise addition, a linear layer, and consequently, an MLP, on $\mathcal{A}$ can be constructed. Following the $C^*$-algebra nets over matrices, we use leaky ReLU for activations.

Table 3: Average test accuracy over five sub-models simultaneously trained on the three datasets. The (fully) noncommutative $C^*$-algebra net outperforms block-diagonal the noncommutative $C^*$-algebra net on Kuzushiji-MNIST and Fashion-MNIST, indicating that the interaction can leverage related tasks.

| Dataset | Commutative | Block-diagonal | Noncommutative |
|---|---|---|---|
| MNIST | $0.956 \pm 0.002$ | $0.969 \pm 0.002$ | $0.970 \pm 0.002$ |
| Kuzushiji-MNIST | $0.745 \pm 0.004$ | $0.778 \pm 0.006$ | $0.796 \pm 0.008$ |
| Fashion-MNIST | $0.768 \pm 0.007$ | $0.807 \pm 0.006$ | $0.822 \pm 0.002$ |

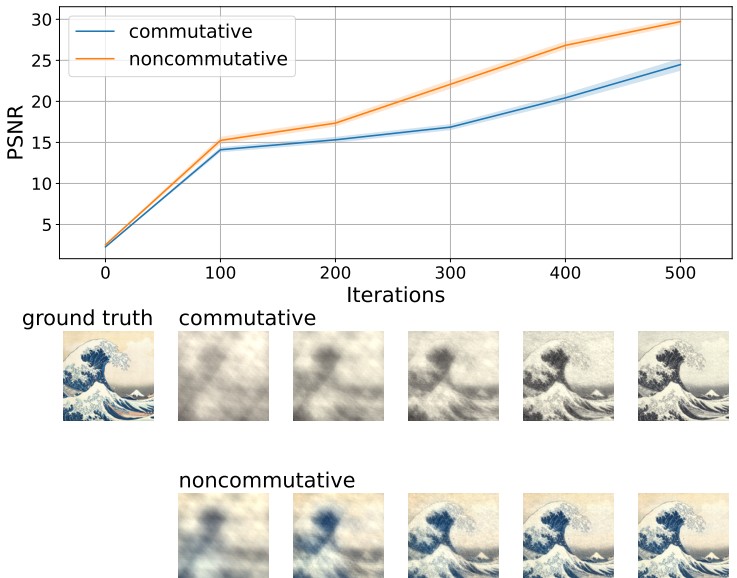

Figure 2: Average PSNR of implicit representations of the image below (top) and reconstructions of the ground truth image at every 100 iterations (bottom). The noncommutative $C^*$-algebra net learns the geometry and colors of the image faster than its commutative counterpart.

One of the simplest tasks for the group $C^*$-algebra nets is to learn permutation-invariant representations, e.g., predicting the sum of given $d$ digits. In this case, $x$ is a function that outputs permutations of features of $d$ digits, and $y(g)$ is a constant function that returns the sum of these digits for all $g \in G$. In this experiment, we use features of MNIST digits of a pre-trained CNN in 32-dimensional vectors. Digits are selected so that their sum is less than 10 to simplify the problem, and the model is trained to classify the sum of given digits using cross-entropy loss. We set the number of layers to 4 and the hidden dimension to 32. For comparison, we prepare permutation-invariant and permutation-equivariant DeepSet models (Zaheer et al., 2017), which adopt special structures to induce permutation invariance or equivariance, containing the same number of parameters as floating-point numbers with the group $C^*$-algebra net.

Table 4 displays the results of the task with various training dataset sizes when $d = 3$. What stands out in the table is that the group $C^*$-algebra net consistently outperforms the DeepSet models by large margins, particularly when the number of training data is limited. Additionally, as shown in Figure 5, the group $C^*$-algebra net converges with much fewer iterations than the DeepSet models. These results suggest that the inductive biases implanted by the product structure in the group $C^*$-algebra net are effective.

## 5   Related works

Applying algebra structures to neural networks has been studied. Quaternions are applied to, for example, spatial transformations, multi-dimensional signals, color images (Nitta, 1995; Arena et al., 1997; Zhu et al., 2018; Gaudet & Maida, 2018). Clifford algebra (or geometric algebra) is a generalization of quaternions,

ground truth

commutative

noncommutative

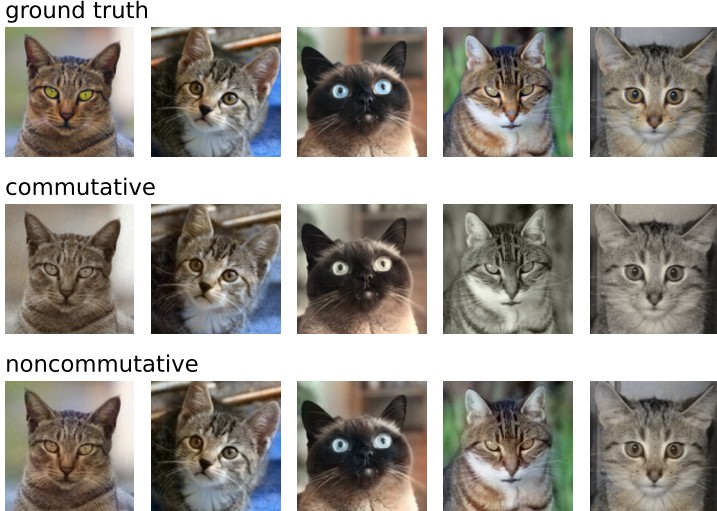

Figure 3: Ground truth images and their implicit representations of commutative and noncommutative $C^*$-algebra nets after 500 iterations of training. The noncommutative $C^*$-algebra net reproduces colors more faithfully.

Ground truth

Commutative

Noncommutative

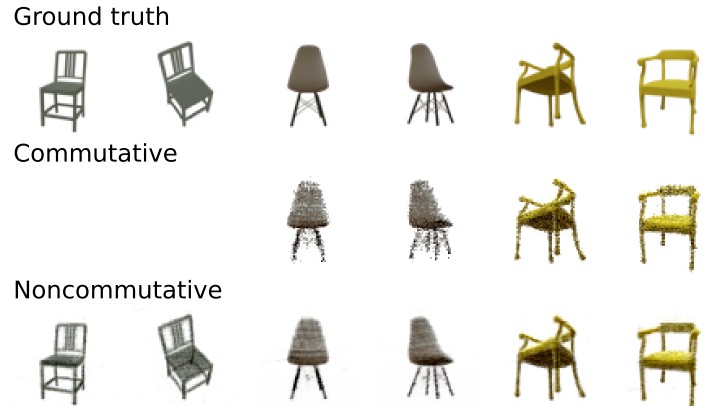

Figure 4: Synthesized views of 3D implicit representations of commutative and noncommutative $C^*$-algebra nets after 5000 iterations of training. The noncommutative $C^*$-algebra net can produce finer details. Note that the commutative $C^*$-algebra net could not synthesize the chair on the left.

and applying Clifford algebra to neural networks has also been investigated to extract geometric structure of data (Rivera-Rovelo et al., 2010; Zang et al., 2022; Brandstetter et al., 2022; Ruhe et al., 2023b;a). Hoffmann et al. (2020) considered neural networks with matrix-valued parameters for the parameter and computational efficiencies and for achieving extensive structures of neural networks. In this section, we discuss relationships and differences with the existing studies of applying algebras to neural networks.

Quaternion is a generalization of complex number. A quaternion is expressed as $a + b\mathrm{i} + c\mathrm{j} + d\mathrm{k}$ for $a, b, c, d \in \mathbb{R}$. Here, i, j, and k are basis elements that satisfy $\mathrm{i}^2\mathrm{j}^2 = \mathrm{k}^2 = -1$, $\mathrm{ij} = -\mathrm{ji} = \mathrm{k}$, $\mathrm{ik} = -\mathrm{ki} = \mathrm{j}$, and $\mathrm{jk} = -\mathrm{kj} = \mathrm{i}$. Nitta (1995); Arena et al. (1997) introduced and analyzed neural networks with quaternion-valued parameters. Since the rotations in the three-dimensional space can be represented with quaternions, they can be applied to control the position of robots (Fortuna et al., 1996). More recently, representing color images using quaternions and analyzing them with a quaternion version of a convolutional neural network was proposed and investigated (Zhu et al., 2018; Gaudet & Maida, 2018).

Table 4: Average test accuracy of an invariant DeepSet model, an equivariant DeepSet model, and a group $C^*$-algebra net on test data of the sum-of-digits task after 100 epochs of training. The group $C^*$-algebra net can learn from fewer data.

| Dataset size | Invariant DeepSet | Equivariant DeepSet | Group $C^*$-algebra net |
|---|---|---|---|
| 1k | $0.408 \pm 0.014$ | $0.571 \pm 0.021$ | $0.783 \pm 0.016$ |
| 5k | $0.731 \pm 0.026$ | $0.811 \pm 0.007$ | $0.922 \pm 0.003$ |
| 10k | $0.867 \pm 0.021$ | $0.836 \pm 0.009$ | $0.943 \pm 0.005$ |
| 50k | $0.933 \pm 0.005$ | $0.862 \pm 0.002$ | $0.971 \pm 0.001$ |

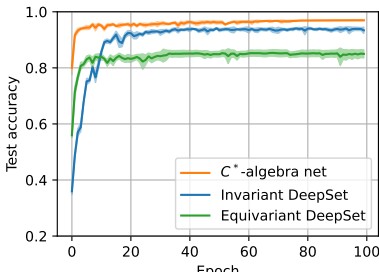

Figure 5: Average test accuracy curves of invariant DeepSet, equivariant DeepSet, and a group $C^*$-algebra net trained on 10k data of the sum-of-digits task. The group $C^*$-algebra net can learn more efficiently and effectively.

Clifford algebra is a generalization of quaternions and enables us to extract the geometric structure of data. It naturally unifies real numbers, vectors, complex numbers, quaternions, exterior algebras, and so on. For a vector space $\mathcal{V}$ equipped with a quadratic form $Q$ and an orthonormal basis $\{e_1, \ldots, e_n\}$ of $\mathcal{V}$, the Clifford algebra is constructed by the product $e_{i_1} \cdots e_{i_k}$ for $1 \leq i_1 < \cdots i_k \leq n$ and $0 \leq k \leq n$. The product structure is defined by $e_i e_j = -e_j e_i$ and $e_i^2 = Q(e_i)$. We have not only the vectors $e_1, \ldots, e_n$, but also the elements whose forms are $e_i e_j$ (bivector), $e_i e_j e_k$ (trivector), and so on. Using these different types of vectors, we can describe data in different fields. Brandstetter et al. (2022); Ruhe et al. (2023b) proposed to apply neural networks with Clifford algebra to solve a partial differential equation that involves different fields by describing the correlation of these fields using Clifford algebra. Group-equivariant networks with Clifford algebra have also been proposed for extracting features that are equivariant under group actions (Ruhe et al., 2023a). Zang et al. (2022) analyzed traffic data with residual networks with Clifford algebra-valued parameters for considering the correlation between them in both spatial and temporal domains. Rivera-Rovelo et al. (2010) approximate the surface of 2D or 3D objects using a network with Clifford algebra. Hoffmann et al. (2020) considered generalizing neural network parameters to matrices. These networks can be effective regarding the parameter size and the computational costs. They also consider the flexibility of the design of the network with matrix-valued parameters.

On the other hand, $C^*$-algebra is a natural generalization of the space of complex numbers. An advantage of considering $C^*$-algebra over other algebras is the straightforward generalization of notions related to neural networks. This is by virtue of the structure of involution, norm, and $C^*$-property. For example, we have a generalization of Hilbert space by means of $C^*$-algebra, which is called Hilbert $C^*$-module (Lance, 1995). Since the input and output spaces are Hilbert spaces, the theory of Hilbert $C^*$-module can be used in analyzing $C^*$-algebra nets. We also have a natural generalization of reproducing kernel Hilbert space (RKHS), which is called reproducing kernel Hilbert $C^*$-module (RKHM) (Hashimoto et al., 2021). RKHM enables us to connect $C^*$-algebra nets with kernel methods (Hashimoto et al., 2023).

From the perspective of the application to neural networks, both $C^*$-algebra and Clifford algebra enable us to induce interactions. Clifford algebra can describe the relationship among data components by using bivectors and trivectors. $C^*$-algebra can also induce the interaction among data components using its product structure. Moreover, it can also induce interaction among sub-models, as we discussed in Section 3.1.1. Our framework also enables us to construct group equivariant neural networks as we discussed in Section 3.1.4.

## 6 Conclusion and Discussion

In this paper, we have generalized the space of neural network parameters to noncommutative $C^*$-algebras. Their rich product structures bring powerful properties to neural networks. For example, a $C^*$-algebra net over nondiagonal matrices enables its sub-models to interact, and a group $C^*$-algebra net learns permutation-equivariant features. We have empirically demonstrated the validity of these properties in various tasks, image classification, neural implicit representation, and sum-of-digits tasks.

A current practical limitation of noncommutative $C^*$-algebra nets is their computational cost. The non-commutative $C^*$-algebra net over matrices used in the experiments requires a quadratic complexity to the number of sub-models for communication, in the same way as the "all-reduce" collective operation in distributed computation. This complexity could be alleviated by, for example, parameter sharing or introducing structures to nondiagonal elements by an analogy between self-attentions and their efficient variants. The group $C^*$-algebra net even costs factorial time complexity to the size of the set, which becomes soon infeasible as the size of the set increases. Such an intensive complexity might be mitigated by representing parameters by parameter invariant/equivariant neural networks, such as DeepSet. Despite such computational burden, introducing noncommutative $C^*$-algebra derives interesting properties otherwise impossible. We leave further investigation on scalability for future research.

An important and interesting future direction is the application of infinite-dimensional $C^*$-algebras. In this paper, we focused mainly on finite-dimensional $C^*$-algebras. We showed that the product structure in $C^*$-algebras is a powerful tool for neural networks, for example, learning with interactions and group equivariance (or invariance) even for the finite-dimensional case. Infinite-dimensional $C^*$-algebra allows us to analyze functional data, such as time-series data and spatial data. We believe that applying infinite dimensional $C^*$-algebra can be an efficient tool to extract information from the data even when the observation is partially missing. Practical applications of our framework to functional data with infinite-dimensional $C^*$-algebras are our future work.

Our framework with noncommutative $C^*$-algebras is general and has a wide range of applications. We believe that our framework opens up a new approach to learning neural networks.

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

# A    Appendix

## A.1    Implementation details

We implemented $C^*$-algebra nets using `JAX` (Bradbury et al., 2018) with `equinox` (Kidger & Garcia, 2021) and `optax` (Babuschkin et al., 2020). For $C^*$-algebra net over matrices, we used the Adam optimizer (Kingma & Ba, 2015) with a learning rate of $1.0 \times 10^{-4}$, except for the 3D NIR experiment, where Adam's initial learning rate was set to $1.0 \times 10^{-3}$. For group-$C^*$-algebra net, we adopted the Adam optimizer with a learning rate of $1.0 \times 10^{-3}$. We set the batch size to 32 in all experiments except for the 2D NIR, where each batch consisted of all pixels, and 3D NIR, where a batch size of 4 was used. Listings 1 and 2 illustrate linear layers of $C^*$-algebra nets using NumPy, equivalent to the implementations used in the experiments in Sections 4.1 and 4.2.

The implementation of 3D neural implicit representation (Section 4.1.2) is based on a simple NeRF-like model and its renderer in Tancik et al. (2021). For training, 25 views of each 3D chair from the ShapeNet dataset (Chang et al., 2015) are adopted with their $64 \times 64$ pixel reference images. The same $C^*$-algebra MLPs with the 2D experiments were used, except for the hyperparameters: the number of layers of four and the hidden dimensional size of 128.

The permutation-invariant DeepSet model used in Section 4.2 processes each data sample with a four-layer MLP with hyperbolic tangent activation, sum-pooling, and a linear classifier. The permutation-equivariant DeepSet model consists of four permutation-equivariant layers with hyperbolic tangent activation, max-pooling, and a linear classifier, following the point cloud classification in (Zaheer et al., 2017). Although we tried leaky ReLU activation as the group $C^*$-algebra net, this setting yielded sub-optimal results in permutation-invariant DeepSet. The hidden dimension of the DeepSet models was set to 96 to match the number of floating-point-number parameters equal to that of the group $C^*$-algebra net.

## A.2    Additional results

Figures 6 and 7 present the additional figures of 2D INRs (Section 4.1.2). Figure 6 is an ukiyo-e counterpart of Figure 3 in the main text. Again, the noncommutative $C^*$-algebra net learns color details faster than the commutative one. Figure 7 shows average PSNR curves over three NIRs of the image initialized with different random states for 5,000 iterations. Although it is not as effective as the beginning stage, the noncommutative $C^*$-algebra net still outperforms the commutative one after the convergence.

```python
import numpy as np

def matrix_valued_linear(weight: Array,
                         bias: Array,
                         input: Array
                         ) -> Array:
    """
    weight: Array of shape {output_dim}x{input_dim}x{dim_matrix}x{dim_matrix}
    bias: Array of shape {output_dim}x{dim_matrix}x{dim_matrix}
    input: Array of shape {input_dim}x{dim_matrix}x{dim_matrix}
    out: Array of shape {output_dim}x{dim_matrix}x{dim_matrix}
    """

    out = []
    for _weight, b in zip(weight, bias):
        tmp = np.zeros(weight.shape[2:])
        for w, i in zip(_weight, input):
            tmp += w @ i + b
        out.append(tmp)
    return np.array(out)
```

Listing 1: Numpy implementation of a linear layer of a $C^*$-algebra net over matrices used in Section 4.1.

ground truth

commutative

noncommutative

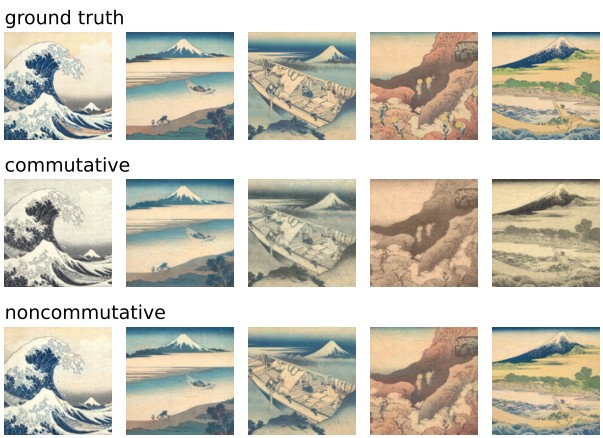

Figure 6: Ground truth images and their implicit representations of commutative and noncommutative $C^*$-algebra nets after 500 iterations of training.

Table 5 and Figure 8 show the additional results of 3D INRs (Section 4.1.2). Table 5 presents the average PSNR of synthesized views. As can be noticed from the synthesized views in Figures 4 and 8, the noncommutative $C^*$-algebra net produces less noisy output, resulting in a higher PSNR.

Figure 9 displays test accuracy curves of the group $C^*$-algebra net and DeepSet models on sub-of-digits task over different learning rates. As Figure 5, which shows the case where the learning rate was 0.001, the group $C^*$-algebra net converges with much fewer iterations than the DeepSet models over a wide range of learning rates, although the proposed model shows unstable results for a large learning rate of 0.01.

```python
import dataclasses
import numpy as np

@dataclasses.dataclass
class Permutation:
    # helper class to handle permutation
    value: np.ndarray

    def inverse(self) -> Permutation:
        return Permutation(self.value.argsort())

    def __mul__(self, perm: Permutation) -> Permutation:
        return Permutation(self.value[perm.value])

    def __eq__(self, other: Permutation) -> bool:
        return np.all(self.value == other.value)

    @staticmethod
    def create_hashtable(set_size: int) -> Array:
        perms = [Permutation(np.array(p)) for p in permutations(range(set_size))]
        map = {v: i for i, v in enumerate(perms)}
        out = []
        for perm in perms:
            out.append([map[perm.inverse() * _perm] for _perm in perms])
        return np.array(out)

def group_linear(weight: Array,
                 bias: Array,
                 input: Array
                 ) -> Array:
    """
    weight: {output_dim}x{input_dim}x{group_size}
    bias: {output_dim}x{group_size}
    input: {input_dim}x{group_size}
    out: {output_dim}x{group_size}
    """

    hashtable = Permutation.create_hashtable(set_size)  # {group_size}x{group_size}
    g = np.arange(hashtable.shape[0])
    out = []
    for _weight in weight:
        tmp0 = []
        for y in g:
            tmp1 = []
            for w, f in zip(_weight, input):
                tmp2 = []
                for x in g:
                    tmp2.append(w[x] * f[hashtable[x, y]])
                tmp1.append(sum(tmp2))
            tmp0.append(sum(tmp1))
        out.append(tmp0)
    return np.array(out) + bias
```

Listing 2: Numpy implementation of a group $C^*$-algebra linear layer used in Section 4.2.

Table 5: Average PSNR over synthesized views. The specified poses of the views are unseen during training.

| Commutative | Noncommutative |
|---|---|
| $18.40 \pm 4.30$ | $25.22 \pm 1.45$ |

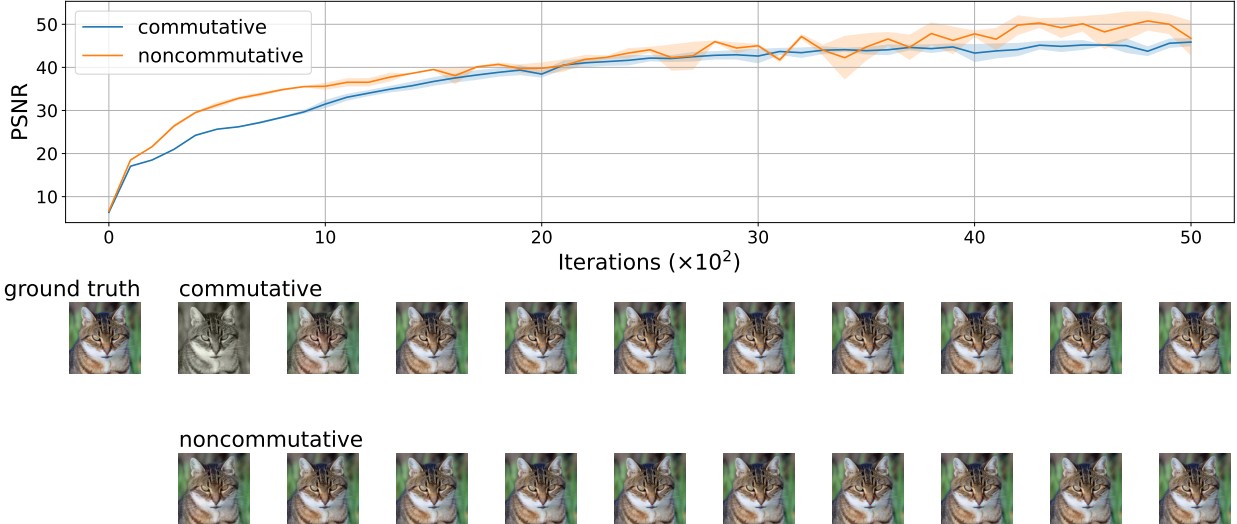

Figure 7: Average PSNR over implicit representations of the image of commutative and noncommutative $C^*$-algebra nets trained on five cat pictures (top) and reconstructions of the ground truth image at every 500 iterations (bottom).

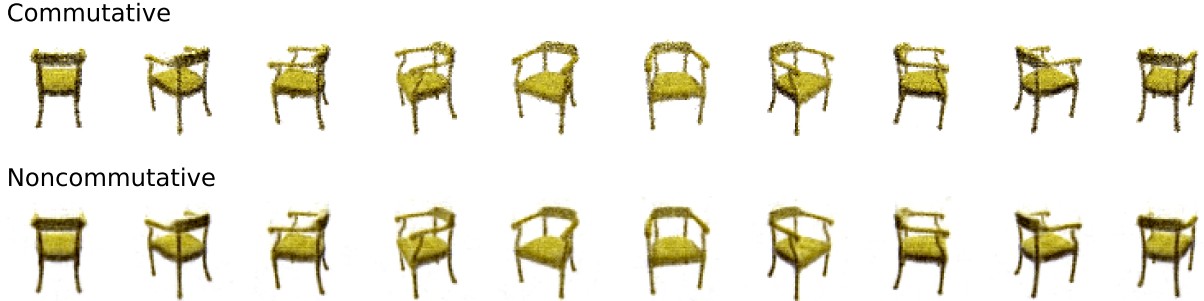

Figure 8: Synthesized views of implicit representations of a chair.

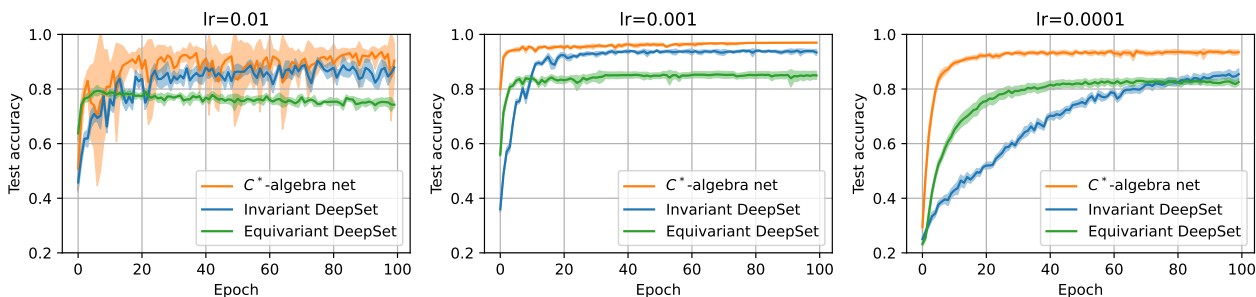

Figure 9: Comparison of test accuracy curves of the group $C^*$-algebra net and DeepSet models over different learning rates.

