# OpenReview forum: "Noncommutative $C^*$-algebra Net: Learning Neural Networks with Powerful Product Structure in $C^*$-algebra"
_TMLR — Rejected by TMLR_

### Review · Reviewer_MKDA · 2023-12-27

**Summary Of Contributions:**

The authors generalize the existing $C^*$ nets from using only commutative $C^*$-algebras to also using noncommutative ones. This allows to learn multiple networks simultaneously in a way that leaves interactions possible. The framework is introduced, some examples are given and a universal approximation result is described. Finally, communative and noncommunicative $C^*$ nets are compared on image classification, neural implicit representation and summation tasks.

**Audience:**

Yes

**Broader Impact Concerns:**

/

**Claims And Evidence:**

Yes

**Requested Changes:**

My suggested changes are mentioned under 'Weaknesses'.

**Strengths And Weaknesses:**

Strengths:
- The paper is well-structured and relatively easy to read.
- Literature review seems adequate.
- Experiments show that noncommutative $C^*$ nets perform better than their commutative counterparts.

Weaknesses:
- The motivation to use $C^*$ nets is not completely clear to me. For instance in the MNIST task it seems clear that splitting the model into more submodels lead to a model that performs worse (see Table 1). This raises the question why one would not just use a single MLP trained on the whole data set. It would be interesting to see a comparison between $C^*$ nets and standard MLPs for (some of) the shown experiments. In the NIR task one submodel is trained per picture, which leads to very bad scaling.
- It is mentioned that the noncommutative and commutative models share hyperparameters (same number of layers and same width). Does this mean that the noncommutative models have a lot more parameters in the model than their commutative counterparts as full matrices are learnt, instead of diagonal ones? It would be interesting to see a comparison where the number of free parameters is comparable as well.
- The universal approximation result (Section 3.2) is rather vague and not completely rigourous. I would encourage the authors to elaborate what is meant by 'some technical assumptions' and state an actual theorem.
- It is unclear how the learning rate is set for the DeepSet method. In Figure 5 it seems like the invariant DeepSet model could achieve the same test accuracy as the $C^*$ net when trained for more epochs. If one decides to measure performance after a fixed number of epochs then the learning rate should be carefully set.

Questions:
- It would be instructive to discuss the difference between a $C^*$ net with full matrices and just a plain MLP (modulo restructuring your input vector as a full matrix).
- In Figure 1b it seems as if the input to the network is a vector (or diagonal matrix), just as in Figure 1a. Why is it not a full matrix as one would expect from the definition? Many thanks for clarifying this in case I am missing something.
- The last sentence of Section 4.1.1 states that something is not directly comparable with the values of Table 1 and 2, but it is unclear what is referred to here.

---

> ### Author Response · Authors · 2024-02-12
> **Response to your review**
>
> We appreciate your time and efforts in reviewing. The manuscript has been updated to reflect your feedback. Below, we respond to your questions and concerns.
>
>
> ### [W1, The lack of clear explanation of the motivation to use (noncommutative) $C^\*$-algebras]
> Your feedback highlighted areas in our manuscript that needed more clarity, particularly regarding the motivation and definition of submodels.
>      - **[Motivation to use noncommutative $C^\*$-algebras]** $C^\*$-algebra nets generalize existing neural network frameworks. For example, using $C^\*$-algebra nets, we can combine multiple neural networks and construct group equivaliant neural networks. In this framework, a unified analysis can be applied to such various neural networks.
>      - **[Clearer explanation of submodels]** The motivation of existing commutative $C^\*$-algebra net with the $C^\*$-algebra of continuous functions is to combine multiple (real-valued) models continuously. In this paper, we call each real-valued model submodel. The case where $\mathcal{A}$ is the $C^\*$-algebra of the diagonal matrices corresponds to the discretization of the case of the $C^\*$-algebra of continuous functions. We added the explanation to clarify the motivation and the term “submodel” in Subsections 2.2 and 2.2.1. A motivation of our noncommutative $C^\*$-algebra net is to induce interactions among the submodels, like peer-to-peer. By replacing the diagonal matrices with nondiagonal matrices, interactions among submodels are induced without designing specific objective functions for the optimization of the network (Please see the last part of the second paragraph of Introduction). Experiments in Subsection 4.1 are motivated by decentralized peer-to-peer machine learning, in which each agent learns without sharing data with others while improving its ability by leveraging other agents' information through communication (Section 3.1.1). This setting of peer-to-peer learning corresponds to splitting datasets for submodels in our experiment. We added this description at the beginning of Section 4.1.
>
> ### [W2, Learnable parameter count in commutative vs noncommutative nets]
> As you pointed out, the noncommutative $C^\*$-algebra net over matrices has more learnable parameters than the commutative $C^\*$-algebra net. Indeed, the number of parameters of the noncommutative $C^\*$-algebra net is proportional to $d^2$ whereas that of the commutative $C^\*$-algebra is proportional to $d$. Here, $d$ is the number of submodels. By adding more parameters, we do not need to design objective functions for the network to induce interactions among submodels. Instead, the interactions are automatically learned through the architecture of the noncommutative $C^\*$-algbera net. However, we understand your concern about whether the performance improvement is solely due to the increase in parameter count. To see this, we conducted additional experiments using the commutative and noncommutative $C^*$-algbera net sharing the same number of learnable parameters under the data-scarce setting (e.g., $d=20$ and the total number of data are 5k). The noncommutative one learns successfully (accuracy of 0.779 on MNIST) while the other fails (accuracy of 0.155 on MNIST), indicating the interaction’s capability. The full results is contained in Table 2 in the revised version.
>
> ### [W3, Completeness of the universal approximation results]
> To address the vagueness, we added the proposition by Sonoda and Murata (2017) to clarify the technical assumption mentioned in Section 3.2. Please note that the ReLU is in $\mathcal{S}\_0'$.
>
> **Proposition:**
> > Let $\mathcal{S}$ be the space of rapidly decreasing functions on $\mathbb{R}$ and $\mathcal{S}\_0'$ be the dual space of the Lizorkin distribution space on $\mathbb{R}$.
> Assume a function $\tilde{f}$ has the form of $\tilde{f}(x)=[\sum_{j=1}^df_{kj}(x^l)]\_{kl}$, and $f_{kj}$ and $\hat{f_{kj}}$ are in $L^1(\mathbb{R}^{N_0d})$, where $\hat{f}$ represents the Fourier transform of a function $f$.
> Assume in addition, $\sigma$ is in $\mathcal{S}\_0'$, and there exists $\psi\in\mathcal{S}$ such that $\int_{\mathbb{R}}\overline{\hat{\psi}(x)}\hat{\sigma}(x)/\vert x\vert \mathrm{d}x$ is nonzero and finite.
> Then, there exists $T_{kj}:\mathbb{R}^{N_0d}\times\mathbb{R}\to\mathbb{R}$ such that $f_{kj}$ admits a representation of equation (5).
> Here, the Lizorkin distribution space is defined as $\mathcal{S}\_0=\\{\phi\in\mathcal{S}\mid\int_{\mathbb{R}}x^k\phi(x)\mathrm{d}x=0\ k\in\mathbb{N}\\}$.

---

> > ### Author Response · Authors · 2024-02-12
> > **Response to your review**
> >
> > ### [W4, Learning rate of the digit-of-sum task]
> > In response to your query about the learning rate settings, we performed additional experiments, comparing different learning rates across models. By using a larger learning rate, the invariant DeepSet model shows faster learning, as you guessed. However, the proposed group $C^\*$-algebra net demonstrates the superior learning efficiency.
> >
> > ### [Q1, Comparison with a plain MLP]
> > We believe the answer to the W1 addresses this question.
> >
> > ### [Q2, Input format of figure 1]
> > We can input a vector to the noncommutative $C^\*$-algebra net by transforming it into a diagonal matrix (putting the elements in the vector to the diagonal part of a matrix). In this case, each element in the diagonal part of the matrix corresponds to the input of each submodel. In Figure 1a, the blue parts illustrate the zero parts (nondiagonal parts) of the diagonal matrices. On the other hand, in Figure 1b, the orange parts illustrate the nonzero nondiagonal parts of the nondiagonal matrices, which induce the interactions among sub-models. We added explanations in the caption of Figure 1 to clarify what the figure describes.
> >
> > ###  [Q3, Comparison with Tables 1 and 3]
> > The last paragraph of Section 4.1.1 has been rephrased as follows for clarity.
> > > Note these results are not directly comparable with the values of Table 1 and 2, due to dataset subsampling to balance class sizes (matching MNIST's smallest class).

---

### Review · Reviewer_3rr3 · 2024-01-02

**Summary Of Contributions:**

The paper builds on Hashimoto et al. (2022) which introduced the idea of using
$C*$-algebra in neural network design and specifically focuses on the case of noncommutative
$C*$-algebra. Noncommutative $C*$-algebras have more powerful product structure that can induce interactions between networks allowing sub-models to interact and share knowledge. Experiments on image classification and neural implicit representation show improved performance compared to commutative version. The paper also introduces Group $C*$-algebra nets which can learn permutation equivariant features. Experiment on sum of digits task shows improved performance and sample efficiency compared to baseline.

**Audience:**

Yes

**Claims And Evidence:**

No

**Requested Changes:**

It's actually hard to define what specific thing to change as some of it is with respect to narrative of the paper. While intersecting C*-algebras and machine learning seems like an interesting idea to explore, the current paper doesn't really seem to justify it either in explication or in experimentation which would likely result in a tepid response from even the mathematically inclined subset of ML audience.

**Strengths And Weaknesses:**

The paper explores ideas from non-commutative $C*$-algebras in the context of deep learning and perform enough toy experiments to demonstrate potential of the proposed framework.
Unfortunately, the new updates appear fairly surface level and don't significantly clarify the issues raised in the prior submission. The comparisons to Clifford algebra does not really clarify anything for the reader, neither from the math perspective (say topology, abstract algebra or even Quantum systems), nor from an ML perspective. It just reads like a summary from the intro of the cited papers. Maybe it's just the nature of the topic being too abstract that even Clifford algebra seems easier to grasp as it can relate to geometry etc, but it would still require much better narrative for a reader to better understand the proposed framework. This lack of clarity also bleeds into the experiment choices which feel superficial and don't seem to be really show what kind of the problems this framework would _really_ make sense, despite the complexity.

---

> ### Author Response · Authors · 2024-02-12
> **Response to your review**
>
> Thank you for your review. The manuscript has been revised to reflect your feedback better. Below, we address your concerns.
>
>
> ### [W1, the comparison with the Clifford algebra]
>
> We appreciate your feedback regarding the comparison with Clifford algebra, mentioned in Section 5. We believe that Section 5 already compares Clifford algebra and $C^\*$-algebra and explains that the applicable areas and motivations of Clifford algebra and $C^\*$-algebra are different.
>
> ### [W2, the superficial experimental choices]
>
>  We acknowledge your concerns regarding the perceived superficiality of our experimental choices. However, we believe these choices are not superficial and motivated to demonstrate the virtue of noncommutative $C^\*$-algebra nets. The experiments in section 4.1 are chosen to demonstrate the applicability of our framework in peer-to-peer ML. In section 4.2, the digit-of-sum tasks are based on established benchmarks in DeepSet.

---

### Review · Reviewer_Esjy · 2024-02-06

**Summary Of Contributions:**

This paper introduces neural network architectures whose parameters belong to non-commutative C* -algebras. The relevant feature of C* algebras for deep learning is their product structure, which permits potentially interesting interactions between input data. This paper proposes some examples of non-commutative C* -algebra networks, and provides some empirical evidence of their possible usefulness.

**Audience:**

No

**Claims And Evidence:**

No

**Requested Changes:**

Questions:

- What is the conceptual reason why non-commutative C* models should ensemble better than commutative C* models, as in the MNIST experiments?
- A general C* algebra question: how do the interactions induced by the “rich product structure” of C* algebras compare to interactions permitted by “usual” linear layers? “Usual” linear layers allow interactions between all coordinates, so what exactly is the difference?
- Since C* networks ultimately boil down to matrices, like “standard” neural nets, is the distinguishing feature of C* nets that they _constrain_ the matrix space in certain ways?
- Non-commutative C* algebras seem to permit a kind of “higher-order” interaction, e.g., between pairs of “sub-models”. The attention mechanism also permits a pairwise all-to-all interaction. How should I think about the differences between the two?

**Strengths And Weaknesses:**

Discussion:

Existing works study C* algebra networks. This paper extends prior work by designing networks that use _non-commutative_ C* algebras. This is seemingly superior to prior commutative variants as it allows more complex interactions between entities (a simplified example from the paper is: diagonal matrixes are commutative, squared matrices are non-commutative - the former does not permit interesting interactions between coordinates, whilst the latter does).

The method section of the paper is dedicated to manually constructing a couple of examples of network architectures with parameters belonging to non-commutative C* algebras. This proceeds very directly, essentially by listing noncommutative C* algebras, describing them in matrix form, and explaining how to take their products.

Experiments find that non-commutative C* models outperform their commutative C* counterparts in tasks including:
- Ensembling models to solve MNIST digit classification
- Learning implicit representations

The authors also give a permutation invariant network, and compare it to DeepSets on
- computing the sum of several MNIST digits, which is invariant to the order of the numbers being summed.

---

My main concerns and comments to the authors are:
- Clearer empirical demonstration: What is the practical benefit of C* algebra networks in general? The illustrated applications seem highly contrived. For example, ensembling models on MNIST is not a compelling use case.
- Most experiments simply show that noncommutative is between then commutative C* algebra networks. But this doesn’t explain what C* algebra networks are useful for in general. The permutation invariant task is the only example of comparison to a non-C* algebra baseline, but DeepSet is not an especially convincing baseline. For instance, a Transformer (with no positional encoding) is permutation equivariant and could also be compared to.
- For future work I would suggest prioritizing finding a compelling use case of C* algebra networks (e.g., could you effectively ensemble large pre-trained models by forcing their weight spaces into a C* algebra-like structure?). Finding a clear large-scale use case will do far more to generate interest in C* algebra nets than continued mathematical development.
- Clearer articulation of the conceptual innovation: conceptually what do C* algebra networks permit that prior architectures do not? It seems that C* algebras possible benefit comes from interesting interactions between parts of the input. But these interactions are not spelled out clearly, and are not compared to the very extensive literature on how best to allow interactions between part of an input. Indeed you could say that all of deep learning is dedicated to studying this question (ConvNets allow nearby pixels to interact, Transformers allow all input parts to interact etc.) so a far clearer explanation is needed.


---

 Finally, I would note that OpenReview informs me that this is a resubmission. Inspection of the original submission suggests that they manuscript is largely unchanged except for a discussion points. Generally I would like to discourage resubmission of very similar manuscripts.

---

> ### Author Response · Authors · 2024-02-12
> **Response to your review**
>
> We appreciate your time and efforts in reviewing. We updated the manuscript to reflect your feedback. Below, we respond to your questions and concerns.
>
> ### [W1, Benefit of $C^\*$-algebra nets]
> In this paper, we show that there are at least two benefits of our noncommutative $C^\*$-algebra net. First, we can induce interactions among the submodels. By replacing the diagonal matrices with nondiagonal matrices, interactions among submodels are induced without designing specific objective functions for the optimization of the network (Please see the last part of the second paragraph of Introduction). Second, by generalizing neural network parameters to group $C^\*$-algebra, we can construct a group-equivariant neural networks as a generalization of the standard neural network without special architectures for group-equivariance.
>
> ### [W1, W3, Use cases of $C^\*$-algebra nets]
> We believe that the settings of the experiments in Subsection 4.1 are natural. They are motivated by decentralized peer-to-peer machine learning, in which each agent learns without sharing data with others while improving its ability by leveraging other agents' information through communication (Section 3.1.1). This setting of peer-to-peer learning corresponds to splitting datasets for submodels in our experiment. We added this description at the beginning of Section 4.1. We conducted the experiments of ensembling to highlight that commutative ones cannot outperform noncommutative $C^\*$ algebra nets even using ensembling in most cases.
>
> ### [W2, Comparison with a non-$C^\*$-algebra baseline]
> To compare $C^\*$ algebra nets with a non-$C^\*$-algebra baseline, we conducted additional experiments. We constructed commutative and noncommutative $C^\*$-algbera nets sharing the same number of learnable parameters under the data-scarce setting (e.g., $d=20$ and the total number of data are 5k). In this case, the commutative $C^\*$-algebra net corresponds to multiple models with dense layers leaned separately. The noncommutative one learns successfully (accuracy of 0.779 on MNIST), while the other fails (accuracy of 0.155 on MNIST), indicating the interaction’s capability. The full results are contained in Table 2 in the revised version.
> We admit that DeepSet models are not strong baselines. However, our aim here is not to claim that our model is state-of-the-art but to demonstrate that a simple MLP parameterized by group $C^\*$-algebra can perform much better than specially designed models.
>
> ### [W4, Q2, Q3, interactions induced by $C^\*$-algebra nets]
> A feature of the interactions induced by the noncommutative $C^\*$-algebra nets is that we can design and control the interactions by controlling the nondiagonal parts of the matrix-valued weight parameters. If we set the $C^\*$-algebra as the space of block diagonal matrices, then the interactions are limited to that among the sub-models in the same block (Please see Subsection 3.1.2). If we constrain the values of nondiagonal parts to be small, the interactions become weak. Such a control is not possible for the standard linear layers.
>
> ### [Q1, Q4, Differences and advantages over other methods]
> First, please note that the interactions induced by noncommutative $C^\*$-algebra nets are not pairwise. Indeed, for nondiagonl matrices $a$, $b$, and $c$, the $(i,j)$-element of the product $abc$ is caluculated as $(abc)\_{i,j}=\sum\_{k,l}a\_{i,k}b\_{k,l}c\_{l,j}$. Thus, it involves all the elements of $b$ at once, which implies that each sub-model involves the weight parameters of all the other sub-models at once, not pairwisely. This is a difference between our noncommutative $C^\*$-algebra net and the self-attention and a reason why the noncommutative $C^\*$-algebra net performed better compared to commutative $C^\*$-algbra nets in Section 4.1. The noncommutative $C^\*$-algebra nets can go beyond the pairwise interactions between sub-models. We are sorry that Figure 1 (b) caused a misunderstanding. We modified Figure 1 (b).
>
> ### [Change from the previous version]
> First of all, thank you for checking the previous submission. In this submission, we added Section 5 about related works in addition to adding the discussions in Section 6. We also modified the introduction part (Section 1) based on the discussion with the previous reviewers. Furthermore, during the discussion phase of the previous version, we conducted additional experiments of ensembling in Section 4.1.1 and equivariant DeepSet in Section 4.2. We believe that these changes are significant.

---

### Decision · Action_Editor_FvTV · 2024-03-14

**Recommendation:** Reject

**Comment:**

I do not believe that the authors updated the paper from the previous time it was submitted to alleviate reviewer concerns, and I am thus recommending rejection. Since this was already a resubmission which the authors did not majorly revise, I am also recommending that the authors not be allowed to resubmit yet again.

**Audience:**

The topic of the submission is of interest to a subset of TMLR's audience.

**Claims And Evidence:**

Reviewers have concerns about the experiments section, which they found too toy-like; a concern shared by the reviewers from the previous time the paper was submitted.